# Assembling covalent organic framework membranes via phase switching for ultrafast molecular transport

Niaz Ali Khan [1,2], Runnan Zhang [1,2,3,4✉], Xiaoyao Wang [1,2], Li Cao[1], Chandra S. Azad [5], Chunyang Fan[1,2], Jinqiu Yuan [1,2], Mengying Long [1,2], Hong Wu [1,3,4,6✉], Mark. A. Olson [7] & Zhongyi Jiang [1,2,3,4✉]

Fabrication of covalent organic framework (COF) membranes for molecular transport has excited highly pragmatic interest as a low energy and cost-effective route for molecular separations. However, currently, most COF membranes are assembled via a one-step procedure in liquid phase(s) by concurrent polymerization and crystallization, which are often accompanied by a loosely packed and less ordered structure. Herein, we propose a two-step procedure via a phase switching strategy, which decouples the polymerization process and the crystallization process to assemble compact and highly crystalline COF membranes. In the pre-assembly step, the mixed monomer solution is casted into a pristine membrane in the liquid phase, along with the completion of polymerization process. In the assembly step, the pristine membrane is transformed into a COF membrane in the vapour phase of solvent and catalyst, along with the completion of crystallization process. Owing to the compact and highly crystalline structure, the resultant COF membranes exhibit an unprecedented permeance (water ≈ 403 L m$^{-2}$ bar$^{-1}$ h$^{-1}$ and acetonitrile ≈ 519 L m$^{-2}$ bar$^{-1}$ h$^{-1}$). Our two-step procedure via phase switching strategy can open up a new avenue to the fabrication of advanced organic crystalline microporous membranes.

[1] Key Laboratory for Green Chemical Technology of Ministry of Education, School of Chemical Engineering and Technology, Tianjin University, 300072 Tianjin, China. [2] Collaborative Innovation Center of Chemical Science and Engineering (Tianjin), 300072 Tianjin, China. [3] Haihe Laboratory of Sustainable Chemical Transformations, 300192 Tianjin, China. [4] Zhejiang Institute of Tianjin University, 315201 Ningbo, Zhejiang, China. [5] Department of Chemistry, Northwestern University, 2145 Sheridan Rd., Evanston, IL 60208, USA. [6] Tianjin Key Laboratory of Membrane Science and Desalination Technology, Tianjin University, 300072 Tianjin, China. [7] Department of Physical and Environmental Sciences, Texas A&M University Corpus Christi, 6300 Ocean Dr., Corpus Christi, TX 78412, USA. ✉email: runnan.zhang@tju.edu.cn; wuhong@tju.edu.cn; zhyjiang@tju.edu.cn

Membrane technology has been envisioned as the disruptive and essential technology to replace the current energy-intensive separation applications[1–5]. Exploring new materials and fabrication strategies for compact, highly ordered microporous membranes is a persistent pursuit[6–10]. Covalent organic frameworks (COFs) are a class of crystalline polymers with tunable and permanent pores, atomically ordered pore structures, and high porosity, constructed by the covalent linkage of organic building blocks (or monomers) based on reticular chemistry[11–14]. The diversity of organic building blocks endows COFs with exceptional structural designability, enabling the customization of the pore size and the functionality at molecular level[15–18]. For instance, choosing appropriate monomers, the COF pores can be manipulated to separate molecules based on size, shape, or charge[19–21]. Similarly, COFs with desired properties can be tailored through linkage modification or linkage transformation by taking advantage of their reversible nature[22–27]. These features make COFs potent materials for the fabrication of advanced membranes for selective molecular transport[28].

COFs are mostly fabricated through a one-step procedure in liquid phase(s) by highly coupled polymerization and crystallization processes[29]. Majority of the reported COF membranes are assembled using this one-step procedure, predominantly in liquid phase using interfacial (oil–water biphasic system) or in-situ solvothermal (monophasic system) methods[30]. In liquids, controlling the concurrent polymerization and crystallization during membrane formation is quite challenging; primarily due to the high surface tension ($2 \times 10^{-2} - 5 \times 10^{-2}\,\mathrm{N\,m^{-1}}$)) and viscosity (0.3–4 cp) of liquids which makes the removal of by-product from the reaction site extremely difficult[31,32]. Consequently, the concentration of reacting monomers is low near the polymerization sites whereas that of byproducts is higher, leading to hindered reaction reversibility[20,33,34]. Moreover, the random movement of monomers/nanoparticles in liquid phase also leads to the formation of loose and low-crystallinity membranes[22]. Recently, we reported that eliminating liquids during the assembly of COF membranes such as at the solid–vapor interface could fabricate compact and highly crystalline membranes[33]. However, the dependence on the melting point of the vapor phase monomer dramatically restricts broad applicability. We envisaged that exploring a two-step procedure instead of one-step procedure to decouple the polymerization reaction process and the crystallization assembly process to the directed evolution of membrane structure, in hopes of achieving some breakthroughs in advanced COF membrane materials.

Herein, we report a two-step procedure to assemble COF membranes via a phase switching strategy. In the first step, i.e., the pre-assembly step, a mixed solution containing aldehyde and amine monomers was casted onto a support and underwent the polymerization process to obtain pristine membranes after solvent evaporation. In the assembly step, the pristine membranes underwent the crystallization process and were assembled into highly crystalline and compact COF membranes in the vapor phase of solvents and catalyst. The thickness of membranes was controlled to ≈150 nm. This phase switching strategy is validated by fabricating two kinds of COF membranes, which exhibit one of the highest permeance during nanofiltration separation.

## Results

### Fabrication and structural characterization of membranes.

The fabrication of COF membranes in this work contains a pre-assembly step and an assembly step. As shown in Fig. 1a, monomers with different spatial configuration i.e., a C-2, 1,4-phenylenediamine (PDA), a C-3, 4,4′,4″-(1,3,5-triazine-2,4,6-triyl) trianiline (TTA) as amine monomers, and a C-3, 1,3,5-triformyl phloroglucinol (TFP) as aldehyde monomer, were chosen. In the first step, i.e., the pre-assembly step, a solution of mixed monomers (aldehyde and amine) was casted onto an Indium Tin Oxide (ITO) coated support to form a pristine polymeric solid membrane following evaporation of the solvent. The temperature was controlled at 60 °C to further exploit the reversibility of the imine linkages. In the second step, i.e., the assembly step, the pristine membrane was transformed into COF membrane in vapor phase containing solvents and catalyst at 145 °C through the linkage rearrangement. Free-standing COF membranes were obtained by etching the ITO layer. As shown in Fig. 2a, d, both COF membranes have high crystallinity as evidenced by the X-ray diffraction (XRD) pattern. The high-resolution transmission electron microscopy (HR-TEM) images and selected area electron diffraction (SAED) patterns further confirm the high crystallinity of the membranes (Fig. 2b, e). The scanning electron microscopy (SEM) images show defect-free surface with controllable thickness to ≈150 nm (Fig. 2c, f).

The membranes in this study are based on β-ketoenamine-linked Schiff base COFs, which are formed by the reactions which exhibit a meta-stable reversible state at low temperature and a more stable irreversible state at high temperature. The initial reversible Schiff base reaction yields an enol form leading to crystalline arrangement which is subsequently converted to keto–enol tautomer form. Moreover, these COFs exhibit excellent chemical stability under harsh acidic and basic conditions[35,36], unlike the boronated COFs, which are highly sensitive to even small amount of moisture and thus difficult to be utilized in aqueous environments[11]. The pre-assembly step was delicately designed and carefully optimized in order to finally obtain highly crystalline and compact membranes in the assembly step and avoid the strong coupling between polymerization and crystallization. The properties of pristine membrane in the pre-assembly step have pronounced effect on the outcome of the COF membranes in the assembly step. In particular, the increase of solvent evaporation temperature in the pre-assembly step is unfavorable for the subsequent assembly step. Fourier-transform infrared spectroscopy (FT-IR) of pristine membranes prepared at temperatures ranging from 35 to 140 °C was recorded (Fig. 3a). The pristine membranes fabricated at 35 and 50 °C still contains amine bands in the range of 3200–3400 cm$^{-1}$. However, at 60 °C or above, no detectable amine bands were observed confirming the consumption of most of the monomers. Interestingly, it was found that the pristine membranes fabricated at 60 °C or below led to COF membranes with high crystallinity in the assembly step (Fig. 3b). In contrast, the pristine membranes fabricated at 80, 120, and 145 °C showed gradual decrease in crystallinity during the assembly step. We assume that at 60 °C, the pristine membranes are still in a reversible state, an important factor in obtaining high crystallinity in the assembly step. An increase in temperature during the pre-assembly step leads the reaction towards an irreversible state, leaving no freedom for linkage reversibility in the assembly step. Therefore, 60 °C at the pre-assembly step was proposed as the transition temperature between reversible and irreversible state.

The transformation from pristine membranes to COF membranes (exemplified by TFP–PDA) via phase switching was evaluated through FT-IR, XRD, X-ray photoelectron spectroscopy (XPS), Brunauer–Emmett–Teller (BET) analysis and SEM, shown in Fig. 4. FT-IR and XRD of TFP–TTA membranes have been shown in Supplementary Figs. 4 and 5. As shown in the FT-IR spectra, new stretching bands at 1248 and 1580 cm$^{-1}$, corresponding to C=N and C=C, respectively, confirmed the formation of the β-ketoenamine linkage (Fig. 4a). The XRD pattern after 3 h exhibited a peak at a lower 2θ value

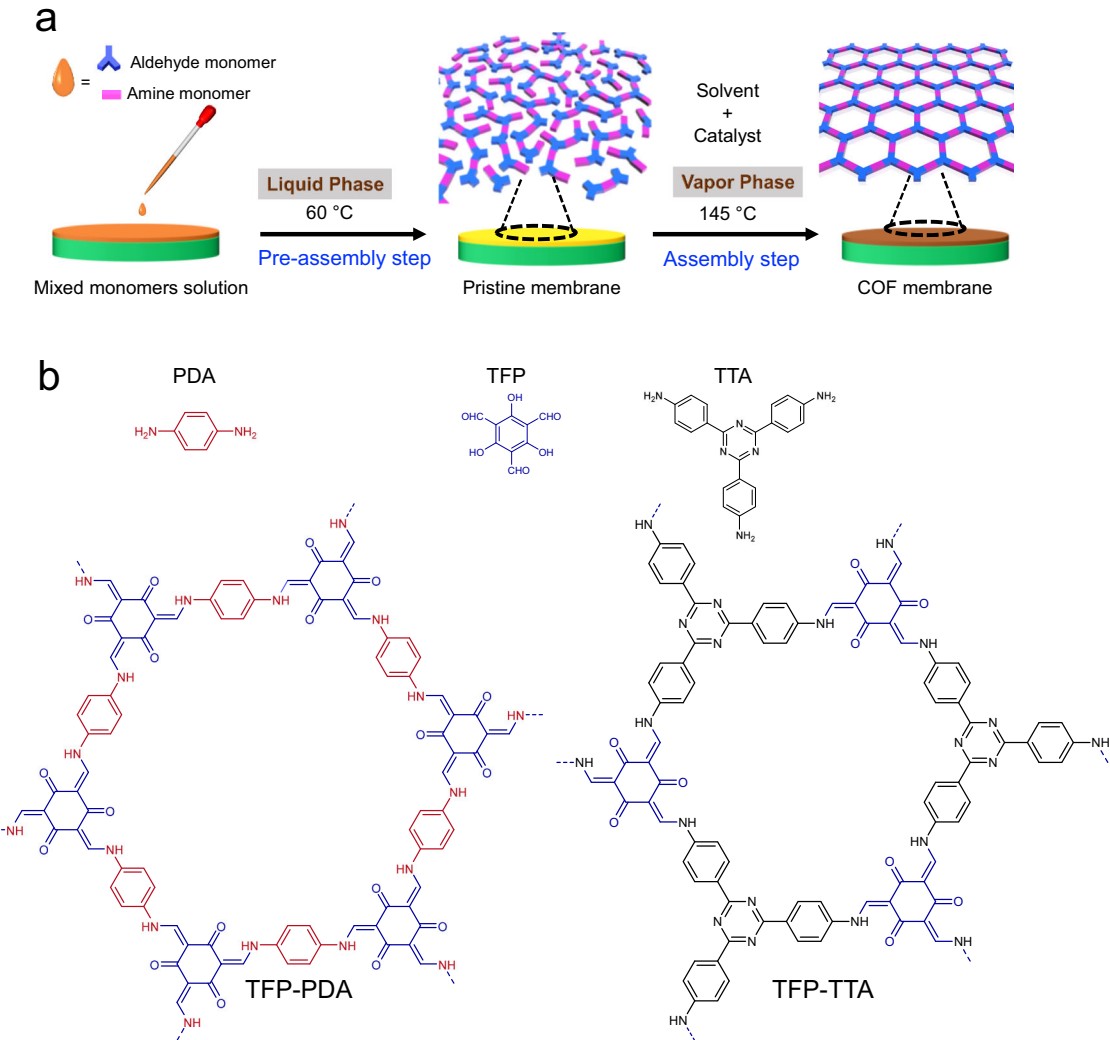

**Fig. 1 Schematic illustration of the membranes fabrication. a** Casting mixed monomer solution on ITO substrate in the pre-assembly step to obtain pristine membrane at 60 °C, subsequently heating the pristine membrane at 145 °C in the assembly step to fabricate COF membranes. **b** Chemical structure of the two COF membranes.

of ≈4.7°, corrsponding to (100) reflection plane, the intensity of which gradually increased with time and reached its maximum value after 18 h of reaction. The second peak at a higher 2$\theta$ value of ≈26.9°, corresponding to (001) reflection plane, is attributed to π–π stacking between two layers (Fig. 4b). The slight deviation from the ideal crystal structure is consistent with the previous literatures[37]. The XPS data in Fig. 4c, d confirms that the linkages in pristine membranes are predominantly in the enol form, whereas the COF exists in $\beta$-ketoenamine form. As shown in Fig. 4e, the TFP–PDA pristine membrane has very low surface area and non-uniform pores. Comparatively, the TFP–PDA COF membrane has high surface area and narrow pore size distribution (Fig. 4f). It is noteworthy that the experimental pore size of the TFP–PDA COF membranes (1.4 nm) calculated from BET is slightly smaller than the theoretical pore size (1.7 nm). This experimental decrease in pore size might be due to the intergrowth of COF particles inside the pores during in-situ healing. The BET surface area and pore size of the TFP–TTA membranes are shown in Supplementary Fig. 6. The morphology of the membrane surface was observed through SEM during the transformation process. As shown in Fig. 4g, the pristine membrane is composed of a thread-like structure which

transforms into a continuous and smooth morphology after 18 h of vapor treatment. The continuous growth of COF membranes indicates that dissolutions and recrystallizations of the COF seeds occurred by the emergence of large seeds due to the fusion of small seeds. These observations are consistent with other crystalline materials such as crystalline salts[38,39], zeolites[40–42], metal-organic frameworks (MOFs)[43,44], and COFs[34]. We also observed that thicker membranes were composed of a continuous membrane towards ITO surface, covered by large particles and threads. These particles and threads could be easily removed using adhesive tape before the etching step (Supplementary Figs. 1–3). By optimizing the monomers concentration, ultra-thin membranes were fabricated by avoiding the growth of particles completely.

The successful fabrication of two COF membranes by using monomers with different symmetries (C2 and C3) and different pore size using our two-step procedure demonstrates its genericity. For comparison, the pre-assembled membrane was treated in the liquid phase containing the same solvents and catalyst at 145 °C. The crystallinity of the as-fabricated membrane was much lower due to the low acidity of AcOH in liquid phase as it exists in the dimer state. Moreover, it was observed that the overall thickness of the membrane obtained in liquid phase was

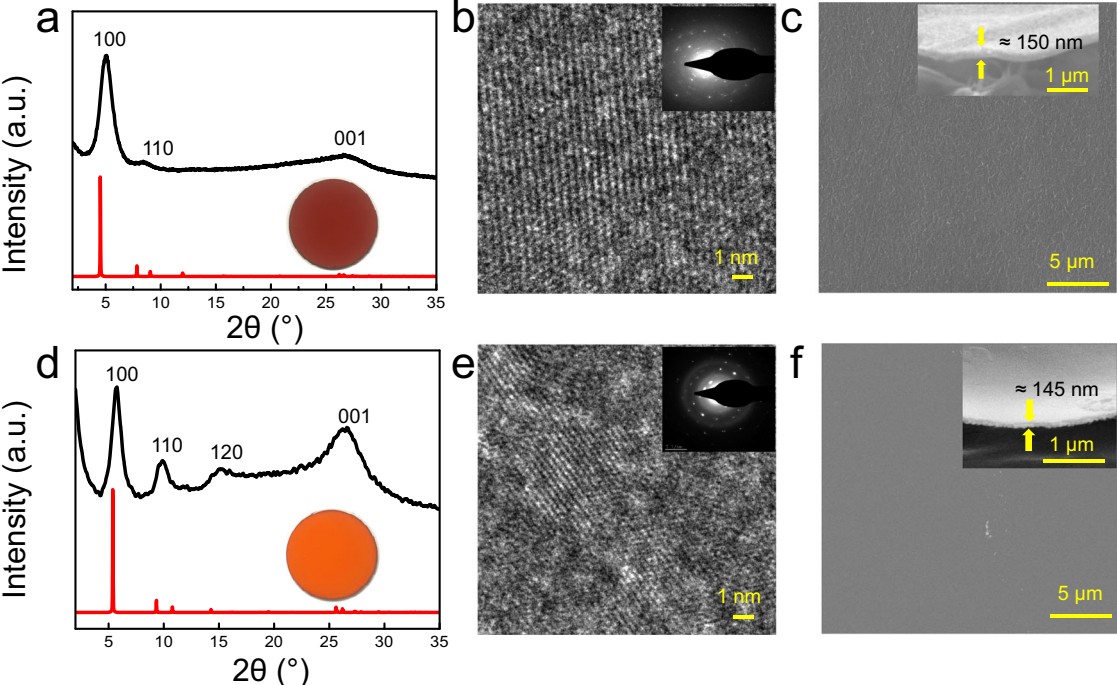

**Fig. 2 Structural characterization. a, d** XRD pattern, experimental (black line) and simulated (red line), digital photos of the COF membranes on the support are also shown. **b, e** HR-TEM images and SAED patterns. **c, f** Surface and cross-section SEM of the TFP–PDA, TFP–TTA, respectively.

thicker than the membranes fabricated by phase switching due to the swelling effect of the liquid phase. However, following removal of the surface particles using adhesive tape, the phase switching membrane turned out to be more densely packed and thicker than membrane obtained in liquid phase (Supplementary Fig. 7). The reason for this can be ascribed to increased random and free movement of particles in the liquid phase, leading to their less dense assembly[34,45]. However, the pre-assembled particles in our work are confined in the solid phase, affording a more appropriate microenvironment for the assembly of a more densely packed membrane.

**Performance evaluation of membranes**. The performance of both membranes for nanofiltration was evaluated. As shown in Fig. 5a, the TFP–PDA COF membranes assembled at 145 °C after 18 h exhibited a water permeance of $403 \pm 4 \, L \, m^{-2} \, h^{-1} \, bar^{-1}$ while the rejection rate of Congo red (CR) as a model dye was over 99%. The water permeance of TFP–PDA COF membranes assembled at 125, 135, and 155 °C was $513 \pm 5$, $444 \pm 5$, and $173 \pm 9 \, L \, m^{-2} \, h^{-1} \, bar^{-1}$, respectively. The CR rejection increased from 78% to 99% when the temperature was increased from 125 to 155 °C (Supplementary Table 1). We propose that at 125 °C, the vapor pressure is not high enough to complete the crystallization process in 18 h, and although the permeance is high, the CR rejection is low due to the incomplete assembly. At 155 °C, the low flux may arise from the partial blockage of pores by intergrown COF particles, also reported in previous literatures[45]. Secondly, treatment at high temperature leads to kinetically stable product rapidly, but with poor orderliness. To confirm this, we characterized TFP–PDA membranes crystallized at 155 °C through XRD and BET. Indeed, the crystallinity and surface area of membranes crystallized at 155 °C is lower than those of membranes crystallized at 145 °C. We assume that growth of COF particles in the pores and low crystallinity at 155 °C render membranes low permeance (Supplementary Fig. 8). The fabrication conditions were further optimized by evaluating the

membranes' performance at a given time vs. temperature (Supplementary Table 1). The permeance of initial polymeric membranes is $800 \pm 10 \, L \, m^{-2} \, h^{-1} \, bar^{-1}$ while the CR rejection rate is at $55 \pm 4\%$. With the elapse of time, the permeance gradually decreased while the CR rejection rate increased. At the low temperature of 125 °C, the decrease in permeance and increase in CR rejection rate becomes slower. However, a sharp and abrupt trend is observed at high temperature (155 °C). At 12 h of reaction time at 155 °C, the permeance is similar to that of membranes obtained at 145 °C in 18 h, but the CR rejection is still at 82%. The reason for the low rejection may be the rapid assembly step at high temperature, resulting in some defects in the membranes[46]. Therefore, membranes fabricated at 145 °C in 18 h in the assembly step were chosen for the subsequent molecular transport performance evaluation. The performance evaluation of TFP–TTA membranes fabricated at various temperatures and at different time is given in Supplementary Table 2.

The TFP–PDA and TFP–TTA COF membranes exhibited high permeance for water and organic solvents (Fig. 5b); especially the TFP–PDA membrane exhibited water ($403 \pm 4 \, L \, m^{-2} \, h^{-1} \, bar^{-1}$) and acetonitrile ($519.6 \, L \, m^{-2} \, h^{-1} \, bar^{-1}$) permeances. This permeance is among the highest ever reported for COF membranes (Supplementary Table 3). To further evaluate the effect of pore size (TFP–PDA = 1.4 nm, TFP–TTA = 1.09 nm), solutions containing different dyes (Alcian blue (AB), $1.25 \times 2.22$ nm; Congo red (CR), $0.73 \times 2.56$ nm; protoporphyrin IX (PPh-IX) 562 g/mol, $1.54 \times 1.45$ nm; methyl blue (MB) $1.74 \times 2.36$ nm; and orange G (OG) $0.85 \times 1.1$ nm) were chosen as model systems. Dyes can form aggregates in aqueous solution at high concentrations[47,48], therefore, an ethanolic solution of PPH-IX was used to confirm the broad applicability of COF membranes for the dyes rejection[1]. As shown in Fig. 5c, both membranes exhibited excellent rejection rates for dyes larger than their corresponding pore size. The rejection rates of larger dyes such as AB, CR, PPH-IX, and MB remained more than 98% for both membranes. The difference in the rejection rates of the smaller dyes become more pronounced. The TFP–TTA membrane, with smaller pores,

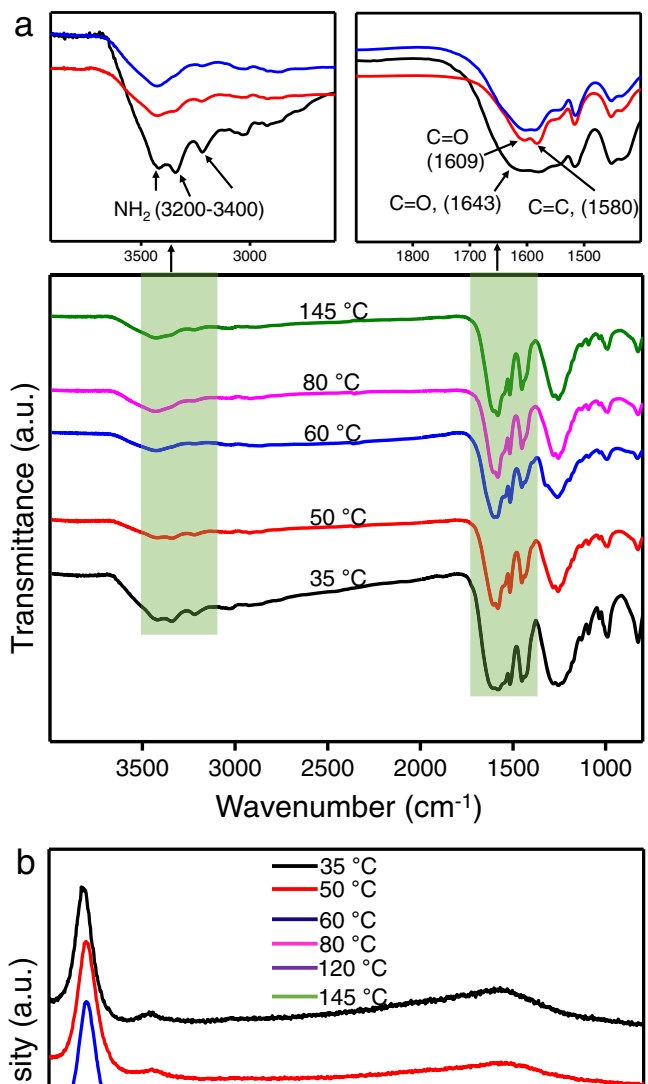

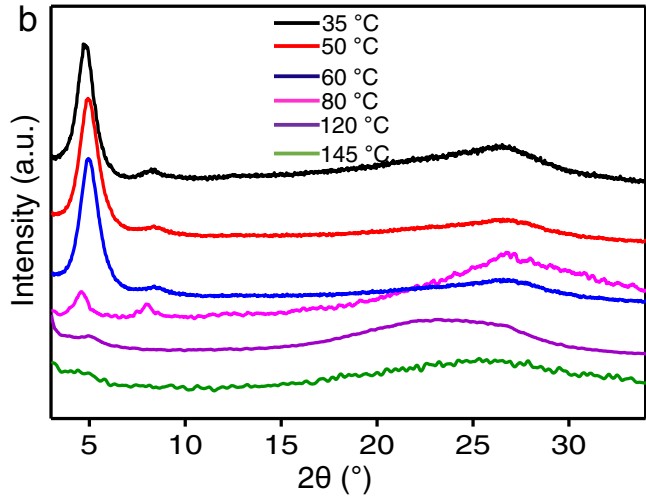

**Fig. 3 Optimization of the temperature at pre-assembly step. a** FT-IR of pristine membrane fabricated at various temperatures (35–145 °C). The disappearance of –NH$_2$ bands between 3200 and 3400 cm$^{-1}$ and the shift of C=O from 1643 to 1609 cm$^{-1}$ has been shown in the insets. **b** XRD of COF membranes (assembled at 145 °C) from TFP–PDA pristine membranes pre-assembled at various temperatures.

exhibited a rejection of 97.8% for OG compared with 79.3% by the TFP-PDA membrane.

Generally, 2D membranes such as of graphene oxide (GO) face the challenge of compactness at high pressure due to their layered structure[49]. Therefore, both membranes were subjected to a variety of pressures ranging from 0.5 to 3 bar. As shown in Fig. 5d, the permeance of both membranes remained unchanged at various pressures, confirming that the pores build the major routes for solvent transport. Similarly, long-term operation stability is a critical requirement for emerging membranes such

as of COFs to compete with state-of-the-art nanofiltration membranes. Therefore, we tested both membranes in long-term operation. As shown in Fig. 5e, both membranes retained high permeance of water (93.7% by TFP–PDA and 96.6% by TFP–TTA) even after 96 h of operation. Similarly, reusability of membranes is another important requirement. The permeance and CR rejection rate of both membranes were evaluated after several cycles of operations (Fig. 5f). Each cycle included filtration of a CR aqueous solution for 30 min followed by filtration of distilled water and subsequent filtration of a CR solution. Due to its larger pore size, TFP–PDA membranes retained 93.4% of initial permeance and 100% of CR rejection while the TFP–TTA membrane retained 85.8% of the initial permeance after 36 cycles. The slight decline in permeance may be ascribed to the clogging of smaller pores by CR dye aggregation. Nevertheless, even after 96 h and 36 cycles of operation, the performance of both membranes was still much higher than the current state-of-the-art membranes.

The reliability of our two-step procedure via phase-switching was further confirmed by the fabrication of COF-LZU1 membranes with defect-free surface and high crystallinity as evident from the XRD data with characteristic peaks similar to the literature[50]. The completion of reaction was confirmed through FT-IR with the disappearance of peaks from the initial monomers and the formation of imine bonding (Supplementary Fig. 9).

## Discussion
In summary, a phase switching strategy was proposed to assemble COF membranes. Unlike the common one-step procedure in liquid phase(s) with concurrent polymerization and crystallization, our two-step procedure of decoupling polymerization and crystallization by switching from liquid phase to vapor phase, can yield highly crystalline and more compact membranes. During the pre-assembly step, the pristine membranes were obtained in the liquid phase, while still at reversible state, which were then transformed into COF membranes in the assembly step in the vapor phase of catalyst and solvent by linkage rearrangement. The broad applicability of this phase switching strategy was confirmed by assembling three COF membranes with different geometries. Owing to the compact and highly crystalline structure, the COF membranes exhibited ultra-high separation performance, as well as superior long-term stability. Our two-step procedure and phase switching strategy may motivate further thinking in how to coordinate the chemical reaction process and the physical assembly process to acquire high-quality COF materials and many other organic crystalline microporous materials.

## Methods
**Pre-assembly step in liquid phase toward pristine membranes**. TFP (2.10 mg, 0.01 mmol), 1,3,5-triformylbenzene (TFB, 1.62 mg, 0.01 mmol), PDA (1.62 mg, 0.015 mmol), TTA (3.54 mg, 0.01 mmol) were dissolved in separate vials in 1 ml DMAc each. Next, the mixed solution containing equal volume of TFP/PDA, TFP/TTA or TFB/PDA was poured on Indium tin oxide (ITO) coated disk; the solvents were evaporated at 60 °C to obtain pristine membranes.

**Assembly step in vapor phase toward COF membranes**. The pristine membranes from the pre-assembly step were placed on top of a glass bottle in a Teflon-lined autoclave. Solvents (oDCB, BuOH) and catalyst (AcOH) in a volumetric ratio of 1:1:0.1 (10 ml in total) were poured on bottom of the Teflon vessel, at least 5 cm below the membrane surface. The whole assembly was heated at 145 °C for 18 h to obtain COF membranes. Free-standing COF membranes were obtained by etching ITO layer through dilute HCl.

**Solvothermal strategy**. The pristine membrane from the pre-assembly step was also treated in the liquid phase containing oDCB, BuOH, and AcOH in a

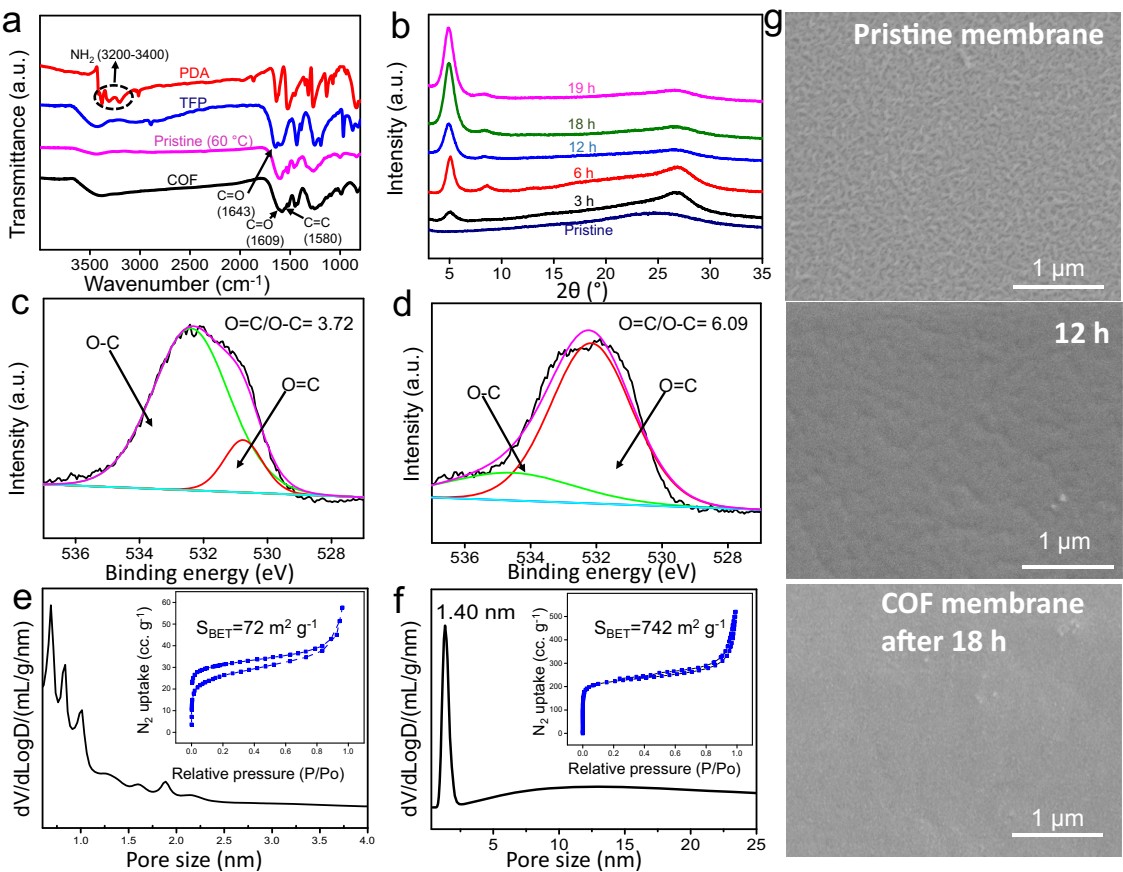

**Fig. 4 Pristine to COF transformation (TFP-PDA). a** FT-IR of monomers, pristine and COF membrane. **b** Time-dependent XRD. **c**, **d** XPS of pristine and COF membranes. **e**, **f** BET of pristine and COF membranes. **g** SEM showing membrane surface change from pristine to COF membranes.

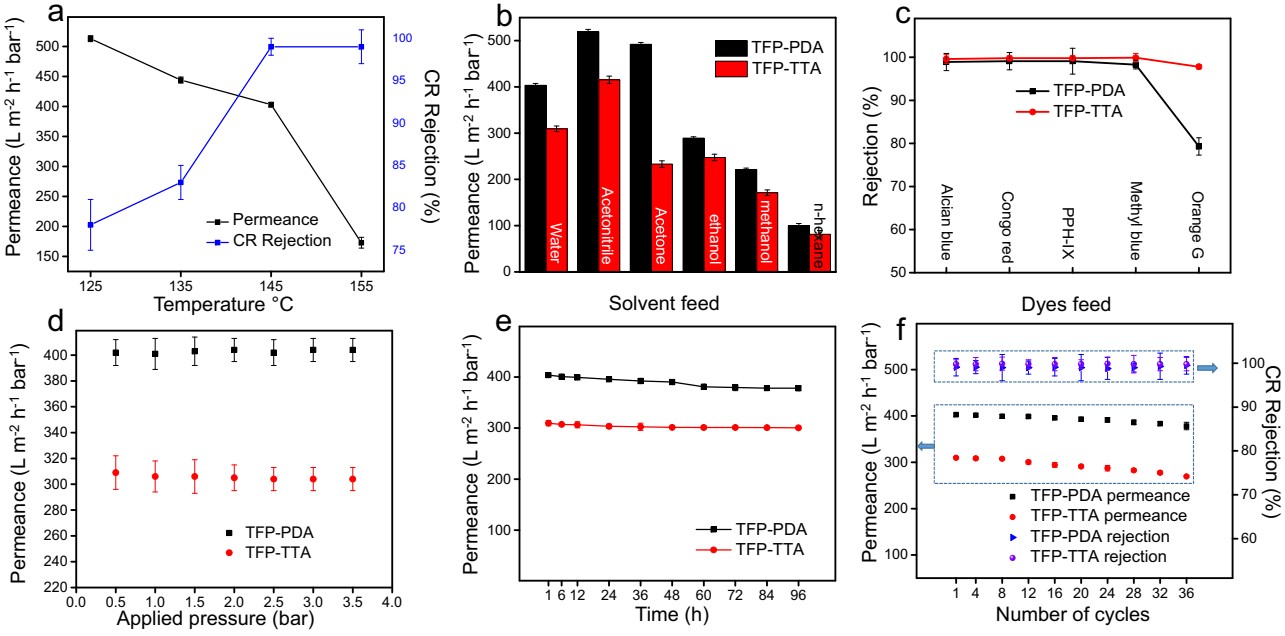

**Fig. 5 Performance evaluation of COF membranes. a** Permeance and CR rejection of membranes fabricated at different temperatures after 18 h. **b** Permeance of various solvents. **c** Rejection (%) of dyes. **d** Permeance at different pressures. **e** Long-term operation. **f** Permeance and CR rejection after specified cycles.

volumetric ratio of 1:1:0.1 (10 ml in total), and heated at 145 °C for 18 h. Free-standing COF membranes were obtained by etching ITO layer through dilute HCl.

**Molecular transport experiments**. A lab made dead-end filtration cell (active area of 4.1 cm²), equipped with magnetic stirrer and nitrogen flow for pressure maintenance, was used for molecular transport. DI water was used for water transport while reagent grade organic solvents were used for organic solvent nanofiltration experiments. All dyes except (PPH-IX) were dissolved in water for molecular separation. PPH-IX was dissolved in ethanol. The concentration of dyes ranged from 50 to 1000 ppm. The performance was evaluated at various cycles of operation. After each cycle, the membrane was washed with water and before next cycle. Equation (1) was used for the calculations of dye rejection:

$$R = \left(1 - \frac{c_p}{c_f}\right) x 100\,\% \tag{1}$$

$c_p$ and $c_f$ are the dyes concentration in permeate and feed solution, respectively.

## Data availability
The authors declare that all data and detailed protocols that support the findings of this study are available within the main manuscript and Supplementary Information files. Additional data from the authors can be provided upon request. Source data are available.

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

## Acknowledgements

This project was supported by Key Research and Development Program of Zhejiang Province, grant No. 2021C03173 (Z. J.), National Natural Science Foundation of China, grant No. 91934302 (Z. J.), 21961142013 (Z. J.), 22008172 (R. Z.), Research Fund for International Young Scientists funded by NSFC, grant No. 21850410457 (N. A. K.), Project funded by China Postdoctoral Science Foundation, grant No. 2020TQ0226 (R. Z.), 2021M692384 (R. Z.), and Program of Introducing Talents of Discipline to Universities, grant No. BP0618007 (Z. J.). We also thank the Haihe Laboratory of Sustainable Chemical Transformations for financial support. We acknowledge the help of Y. Chao during figures drawing.

## Author contributions

N.A.K. designed the project and wrote the initial draft. Corresponding authors, i.e., R.Z., H.W., and Z.J. supervised the project. Z.J. mentored the project during its execution and revised the manuscript. X.W. and L.C. helped during the experiments. J.Y., M.L. performed some parts of the characterization. M.A.O. and C.S.A. gave their valuable theoretical suggestions. X.W. also simulated the XRD patterns of the COF membranes. All authors communicated actively during the writing of the manuscript.

## Competing interests

The authors declare no competing interests.
