## [Peer Review File · Nature Communications]

Assembling covalent organic framework membranes via phase switching for ultrafast molecular transportReviewers' Comments:

Reviewer #1:

Remarks to the Author:

In this manuscript, the authors have reported a two-step procedure to assemble COF membrane via a phase switching strategy. Accordingly, two COFs-based membranes were prepared and well-characterized. In addition, the obtained TFP-PDA COF membrane exhibited a water permeance of $402 \pm 3 \text{ L m}^{-2} \text{ h}^{-1} \text{ bar}^{-1}$ and a good rejection rate (99%) toward organic dyes (Congo red). Given the interesting applications of porous membranes, I would like to recommend the publication on Nature Communications after addressing the questions stated below:

- 1) The pristine membrane was obtained by evaporation of the solvent at 60 °C. However, the monomers were dissolved in DMAc, which has a very high boiling point (> 160 °C). Some explanation should be added here.
- 2) It seems that there is a very small deviation between the simulated and experimental PXRD, such as the 100 peak. Is this possible to have slipped AA structure?
- 3) The authors assumed that the pristine membrane at 60 °C in a reversible state. However, from the FT-IR spectrum, there is no imine bonds, which is different from the presence of amine groups at low temperature. The author should provide more solid evidence to prove their hypothesis. Otherwise, the mechanism for this two-step procedure needs to be reconsidered.
- 4) After finishing the first step, the obtained pristine membrane was further being placed at 145 °C to finish the linkage rearrangement. However, at this temperature, the stable linkage formed after rearrangement seems not good for the morphology repairing of the membrane. How did the membrane become to be more compact and defect free? This also need more discussions.
- 5) Both the thickness of the membranes is about 150 nm. Did the author try to obtain other membrane with different thickness to demonstrate the controllability? The permeance of the membrane may also be improved by optimizing the thickness.
- 6) How did the authors test the gas adsorption of the membrane? More details can be put into the experimental section to compare the difference in porosity between the membrane and the COF powder.

Reviewer #2:

Remarks to the Author:

The paper by Khan et al. reports a 2-step procedure named "phase switching strategy" to achieve greater crystallization. This strategy allows the amine and aldehyde monomers to be cast on solid support and form membranes through solvent evaporation at 60 °C. The crystallization was delayed to the second step, where a high temperature (145 °C) was applied to initiate linkage rearrangement to form a COF membrane. This method seems to be similar to the "vapor-assisted conversion" (J. Am. Chem. Soc. 2015, 137, 3, 1016–1019) and "vapor-assisted solid-state approach" (J. Mater. Chem. A, 2014, 2, 8201–8204) that have been used for crystalline COF membrane synthesis, as well as "steam-assisted conversion" (Angew. Chem. Int. Ed., 2010, 50, 3, 672–675) and "dry-gel conversion" (Thin Solid Films, 2013, 529, 327–332) methods used for other materials. These methods share the same idea of molecular reordering in vaporized solvents that act as structure-directing agents. Hence, the authors may want to clarify the differences between the present method and the reported ones.

The materials used in this paper have been reported in the authors' previous paper (J. Am. Chem. Soc. 2020, 142, 31, 13450–13458) and that of Dey et al. (J. Am. Chem. Soc. 2017, 139, 37, 13083–13091). The TFP-PDA membrane in the previous paper has shown slightly better permeability ($411 \text{ L m}^{-2} \text{ h}^{-1} \text{ bar}^{-1}$ for water and $\sim 583 \text{ L m}^{-2} \text{ h}^{-1} \text{ bar}^{-1}$ for acetonitrile) compared to the present membrane. This may be due to the thicker membranes. Therefore, despite having one of the highest reported permeabilities, this current method does not contribute to a more superior membrane. Referring to Supplementary Table 1, the optimum data is 402 LMHbar with 99 % CR rejection (taken from 145 °C at 18 hours of reaction). This data is inconsistent with the general trend. In fact, we see

an unnatural rebound of the permeability at 18 hours. The authors may wish to explain the mechanism behind increased rejection from 83 % (15 hours) to 99 % while maintaining the permeability at ~ 401 LMH/bar. The optimization table for TFP-TTA is not provided anywhere in the paper.

The authors may wish to explain the mechanism of the high rejection efficiency of Congo Red, which has a size that is small enough (~ 0.73 nm) to permeate through the large pores of TFP-PDA (1.4 nm). Also, the authors may consider giving a more detailed explanation of the low rejection and permeability at 155 °C as reference 38 does not seem to provide the information.

In general, this work is not particularly novel and presents a very similar structure as the authors' previous paper. It may not be suitable for Nature Communications.

Reviewer #3:

Remarks to the Author:

In this work, the authors report a phase-switching strategy to fabricate covalent organic framework (COF) membranes through a two-step procedure. The first step in liquid phase is primarily responsible for polymerization and the second step in vapor phase is primarily responsible for crystallization, which delicately decouples polymerization process and crystallization process. The two-step procedure has some distinct superiority to the commonly used one-step procedure in fabricating defect-free and ultrathin membranes. Using phase-switching strategy, highly crystalline, more compact and continuous membranes were fabricated. Accordingly, ultrafast molecular transport was realized, as manifested by water permeance of $402 \text{ L m}^{-2} \text{ h}^{-1} \text{ bar}^{-1}$ and acetonitrile permeance of $519 \text{ L m}^{-2} \text{ h}^{-1} \text{ bar}^{-1}$, which is among the ever-reported highest separation performance for COF membranes. The phase-switching strategy was validated by fabricating two kinds of highly crystalline COF membranes with different pore apertures. The strategy in this work represents a striking breakthrough in the precise construction of advanced COF membranes. This paper is meticulously organized and elegantly written and I read it with great joy. I firmly believe this interesting and innovative work will be highly appreciated by broad scientific communities in chemistry, materials science and membrane technology. Overall, this work is highly suitable for publication in Nature Communications after addressing the following minor revisions.

1. The authors employed the β -ketoenamine-linked Schiff base COF in their work. More explanations may be required for the readers to more clearly understand why the authors preferred this type of COF over the other types?
2. Can the phase-switching strategy be applied to other kinds of COF membranes in principle?
3. Why did the authors choose DMAc as the solvent in the first step? Can the authors outline the major criteria in choosing the solvent?
4. Have the authors tested the rejection performance of dyes in organic solvents?

Some mistakes:

- (1) The scale bar of cross-section SEM in Fig. 2c, f should be corrected.
- (2) In line 125, XRD was written twice.

REVIEWER COMMENTS

Reviewer #1 (Remarks to the Author):

In this manuscript, the authors have reported a two-step procedure to assemble COF membrane via a phase switching strategy. Accordingly, two COFs-based membranes were prepared and well-characterized. In addition, the obtained TFP-PDA COF membrane exhibited a water permeance of $402 \pm 3 \text{ L m}^{-2} \text{ h}^{-1} \text{ bar}^{-1}$ and a good rejection rate (99%) toward organic dyes (Congo red). Given the interesting applications of porous membranes, I would like to recommend the publication on Nature Communications after addressing the questions stated below:

Reply:

We would like to thank the reviewer for these highly positive remarks and valuable guidance.

1) The pristine membrane was obtained by evaporation of the solvent at 60 °C. However, the monomers were dissolved in DMAc, which has a very high boiling point (> 160 °C). Some explanation should be added here.

Reply:

We would like to thank the reviewer for this valuable guidance. The major reasons for choosing DMAc are the high solubility of amine and aldehyde monomers in DMAc as well as the appropriately slow rate of evaporation for completion of the polymerization in the first step. Due to the monomer's high solubility (TFP \approx 23mg/ml, PDA \approx 25mg/ml and TTA \approx 17 mg/ml), a low volume of DMAc is required in the first step. Secondly, due to the relatively high boiling point of DMAc, it evaporates slowly at 60 °C allowing the completion of polymerization. In our experiments, we observed that 2 ml of DMAc requires about 25 minutes for its evaporation at 60 °C. When a low boiling point solvent such as dichloromethane is used, the solvent evaporates very fast leaving no sufficient time for the completion of the polymerization in the first step. We have added this text to the revised supplementary Information file.

“DMAc was chosen in this step due to high solubility of amine and aldehyde monomers in DMAc which reduces reaction volumes (TFP \approx 23mg/ml, PDA \approx 25mg/ml and TTA \approx 17 mg/ml) and relatively high boiling point of DMAc, which wins sufficient evaporation time for the completion of the polymerization at this step.”

2) It seems that there is a very small deviation between the simulated and experimental PXRD, such as the 100 peak. Is this possible to have slipped AA structure?

Reply:

We thank the reviewer for this valuable observation. Indeed, there is a slight deviation between the experimental and simulated peaks. This deviation is consistent with previous literature (*Angew. Chem. Int. Ed.*, 2021, **60**, 19797-19803). Ideally, an AA stacking of COF layers will result in a membrane with perfectly aligned vertical pores that will afford solvent permeance manifolds higher than what is currently reported. However, in our work and reported literature, there exists a big chance for AB stacking, rendering slight deviation from the simulated pattern. Further exploration is required to achieve perfect AA stacking in future work. We have added relevant discussion on the matter in the revised manuscript.

“is attributed to π - π stacking between two layers (Fig. 4b). The slight deviation from the ideal crystal structure is consistent with the previous literatures¹.”

3) The authors assumed that the pristine membrane at 60 °C in a reversible state. However, from the FT-IR spectrum, there is no imine bonds, which is different from the presence of amine groups at low temperature. The author should provide more solid evidence to prove their hypothesis. Otherwise, the mechanism for this two-step procedure needs to be reconsidered.

Reply:

Again, we thank the reviewer for this observation and guidance. The appearance of imine peaks in the FT-IR are masked due to it overlapping with the strong C=C and C=O peaks in the same region of the spectra. The FT-IR peaks of the imine lie in the region of 1615-1630 cm^{-1} with low intensities (*J. Mol. Struct.*, 2020, **1214**, 128150 (2020)). However, in the COFs reported in this work, the same region is occupied by the peaks of C=C and C=O (1605-1630 cm^{-1}) with higher intensities. The absence of amine bands at $\approx 3300 \text{ cm}^{-1}$ of the amine monomer and the shifting of carbonyl band from 1643 cm^{-1} to 1609 cm^{-1} of the aldehyde monomer can indirectly prove the formation of imine formation.

4) After finishing the first step, the obtained pristine membrane was further being placed at 145 °C to finish the linkage rearrangement. However, at this temperature, the stable linkage formed after rearrangement seems not good for the morphology repairing of the membrane. How did the membrane become to be more compact and defect free? This also need more discussions.

Reply:

We thank the reviewer for this beneficial advice. Generally, the phenomena of dissolution and recrystallization commonly exist in many crystalline materials such as MOFs (*Chem. Commun.*, 2020, **56**, 1960-1963; *J. Power Sources*, 2021, **494**, 229733), zeolites (*Microporous Mesoporous Mater.*, 2007, **102**, 80-85; *J. Solid State Chem.*, 2013, **200**, 179-188; *J. Mater. Chem. A.*, 2020, **8**, 13710-13717) and other materials (*Phys. Met. Metallogr.*, 2020, **121**, 1258-1265; *Clay Miner.*, 2012, **47**, 373-390). We observed that the pristine membrane was composed of tight threads of small grains. With the increase of vapor treatment time, the membranes morphology turned smoother and defect-free. From the SEM observations, the progress is marked with the transformation in Fig. 4g. We suppose that dissolution and recrystallization also occur in this work at high temperature where small COF seeds/grains fuse together to create larger seeds/grains. Such fusion can heal the grain-boundary defects during the whole process, yielding defect-free membranes. Moreover, the presence of hydroxyl groups in these COFs can stabilize membranes morphology due to interlayer molecular hydrogen bonding. We have added this description to the revised manuscript.

“morphology after 18 h of vapor treatment. The continuous growth of COF membranes indicates that dissolutions and recrystallizations of the COF seeds occurred by the emergence of large seeds due to the fusion of small seeds. These observations are consistent with other crystalline materials such as crystalline salts^{2, 3}, zeolites⁴⁻⁶, metal-organic frameworks (MOFs)^{7, 8} and COFs⁹”

5) Both the thickness of the membranes is about 150 nm. Did the author try to obtain other membrane with different thickness to demonstrate the controllability? The permeance of the membrane may also be improved by optimizing the thickness.

Reply:

We would like to thank the reviewer for this helpful insight. In the initial experiments during this work, we observed that the membranes had two distinct layers, the top layer composed of large particles/threads and the bottom layer composed of defect free membrane, as shown in supplementary Fig. 1 and 2. Irrespective of the top layer's thickness, the bottom layer remained at \approx 150 nm after 18 h at 145 °C. We optimized the conditions to reduce the top layer by lowering monomer's concentration until no particles were observed. The permeance was tested for each membrane and we observed a similar permeance for all membranes. We deduce that the top layer does not take part in the permeance and only the bottom layer is responsible for the permeance. In

this work, we also observed that membranes below 140 nm could not be employed for performance evaluation due to low mechanical strength. We have highlighted this part in the revised manuscript.

“We also observed that thicker membranes were composed of a continuous membrane towards ITO surface, covered by large particles and threads. These particles and threads could be easily removed using adhesive tape before the etching step (Supplementary Fig. 1-3). By optimizing the monomers concentration, ultra-thin membranes were fabricated by avoiding the growth of particles completely.”

6) How did the authors test the gas adsorption of the membrane? More details can be put into the experimental section to compare the difference in porosity between the membrane and the COF powder.

Reply:

We thank the reviewer again for this valuable insight and advice. The porosity measurements were done by first fabricating several membranes and grinding them into powders. We have included the following relevant descriptions into the experimental section of the revised supplementary information file.

“N₂ adsorption/desorption isotherms were recorded on PS2-1055-B gas adsorption analyzers at 77 K using a liquid nitrogen bath. Several membranes were first fabricated and then grinded for porosity evaluation. All the membrane samples were degassed at 120 °C for 6 h under vacuum before N₂ analysis.”

Reviewer #2 (Remarks to the Author):

The paper by Khan et al. reports a 2-step procedure named "phase switching strategy" to achieve greater crystallization. This strategy allows the amine and aldehyde monomers to be cast on solid support and form membranes through solvent evaporation at 60 °C. The crystallization was delayed to the second step, where a high temperature (145 °C) was applied to initiate linkage rearrangement to form a COF membrane.

This method seems to be similar to the “vapor-assisted conversion” (J. Am. Chem. Soc. 2015, 137, 3, 1016–1019) and “vapor-assisted solid-state approach” (J. Mater. Chem. A, 2014, 2, 8201–8204) that

have been used for crystalline COF membrane synthesis, as well as “steam-assisted conversion” (*Angew. Chem. Int. Ed.*, 2010, 50, 3, 672–675) and “dry-gel conversion” (*Thin Solid Films*, 2013, 529, 327–332) methods used for other materials. These methods share the same idea of molecular reordering in vaporized solvents that act as structure-directing agents.

Hence, the authors may want to clarify the differences between the present method and the reported ones.

Reply:

Firstly, we would like to thank this reviewer for the important and valuable input. The strategy/method presented in this work has some distinct innovative features which have never been reported in literature: (I) decoupling of polymerization and crystallization steps, and optimizing both steps independently; (II) using vapors instead of liquid for efficient and fast crystallization; and more importantly, (III) the strategy can obtain highly crystalline COF membranes. The reported methods mentioned by the reviewer are all about the synthesis of powders or thick membranes with visible defects observable in their SEM images, which cannot be used for molecular separations. Moreover, these literatures (*J. Am. Chem. Soc.*, 2015, **137**, 3, 1016-1019, *Angew. Chem. Int. Ed.*, 2010, **50**, 3, 672-675 and *Thin Solid Films*, 2013, **529**, 327-332) differ significantly from our strategy in that they are all based on a one-step procedure, i.e., concurrent polymerization and crystallization, not a two-step process as in this work. In the report of *J. Mater. Chem. A*, **2014**, 2, 8201-8204, the vapors were only used to improve the crystallinity of pre-synthesized COF powders (not membranes) through a mechanochemical method.

Furthermore, the methods for synthesizing the COF powders cannot be directly and easily applied to COF membranes because molecular separation membranes require the homogeneous dispersion and appropriate linkage/connection of the powdered materials to avoid the formation of defects, which are not a major concern in powder synthesis. The films fabricated at room temperature in the JACS paper (*J. Am. Chem. Soc.*, 2015, **137**, 3, 1016-1019) and (*Thin Solid Films*, 2013, **529**, 327-332) contains abundant sub-micron defects, as observed in the SEM image, which are not suitable for applications in the field of molecular separations. These defects could not be healed under the conditions using their method.

Selected figures about the reported methods and our method.

Table outlining the comparison of the reported methods in question versus our method.

Reference	Method	Two-steps	Clear boundary between two steps	Membrane structures	Suitable for molecular separations
J. Am. Chem. Soc. 2015, 137, 3, 1016–	Spreading mixed monomer solution on substrate, subsequent processing in vapors	No	No	Several μm thick, large pores	No

1019					
J. Mater. Chem. A, 2014,2, 8201–8204	Mechanochemical synthesis of powder and subsequent processing in vapors	No	No	No	No
Angew. Chem. Int. Ed., 2010, 50, 3, 672–675	Not COF Concurrent reaction and crystallization in steam	No	No	No	No
Thin Solid Films, 2013, 529, 327–332	Not COF Concurrent reaction and crystallization in steam	No	No	Several μm thick, large pores	No
This work	Completion of polymerization in the first step with subsequent crystallization in the second step	Yes	Yes	Yes (≈ 150 nm)	Yes

The materials used in this paper have been reported in the authors' previous paper (J. Am. Chem. Soc. 2020, 142, 31, 13450–13458) and that of Dey et al. (J. Am. Chem. Soc. 2017, 139, 37, 13083–13091). The TFP-PDA membrane in the previous paper has shown slightly better permeability ($411 \text{ L m}^{-2} \text{ h}^{-1} \text{ bar}^{-1}$ for water and $\sim 583 \text{ L m}^{-2} \text{ h}^{-1} \text{ bar}^{-1}$ for acetonitrile) compared to the present membrane. This may be due to the thicker membranes. Therefore, despite having one of the highest reported permeabilities, this current method does not contribute to a more superior membrane.

Reply:

We thank the reviewer for this important observation. In our current work, we reported TFP-PDA and TFP-TTA as model COF membranes to validate our new fabrication strategy. In the literature of *J. Am. Chem. Soc.*, 2017, **139**, 37, 13083-13091, a TFP-TTA membrane was prepared along with a series of other COF membranes using liquid-liquid interfacial polymerization, the membrane preparation took 3 days (72h), but the separation performance of TFP-TTA membranes was not available. In their work, Tp-BPy membrane exhibited the highest water ($211 \text{ L m}^{-2} \text{ h}^{-1} \text{ bar}^{-1}$) and acetonitrile ($339 \text{ L m}^{-2} \text{ h}^{-1} \text{ bar}^{-1}$) permeance, which was in fact lower compared to this work. The method reported in this literature differs greatly from our current strategy. The liquid-liquid interface is a highly confined space, the membranes were prone to breakage into pieces, even with a slight disturbance. The processing time was long (several days), and the monomers face potential solubility issues, i.e., requirement for monomers to be soluble/insoluble in the respective immiscible

solvents to achieve a distinct interface. Moreover, the reaction is difficult to control at the interface, as the amine monomer can diffuse into organic solvent due to its solubility in organic solvent, resulting in polymerization away from the interface yielding thick membranes (several µm). In our previous work of *J. Am. Chem. Soc.*, 2020, **142**, 31, 13450-13458, the TFP monomer was grown on the functionalized $SiO_2$ support, whereas the PDA monomer was evaporated to react with the TFP. The method relied significantly on the melting point of the evaporating monomer. Therefore, monomers with high melting point such as of TTA (≈365 °C) used in the current work cannot be evaporated at 150 °C and therefore will not result in any COF membranes. Secondly, the solid-vapor method in our previous work is an equivalent of chemical vapor deposition method employed for the fabrication of films of other materials such as of graphene oxide etc. Indeed, the current work has slightly lower permeance than our previous work using the same COF, but in the current work, we have focused greatly on the development of a more easily employed approach which can utilize a much broader chemical space, allowing for much more monomers to be employed as building units for COF membrane fabrication, free of the restriction of melting/boiling point. Further exploration of this strategy in the future may yield high permeance.

Referring to Supplementary Table 1, the optimum data is 402 LMHbar with 99 % CR rejection (taken from 145 ˚C at 18 hours of reaction). This data is inconsistent with the general trend. In fact, we see an unnatural rebound of the permeability at 18 hours. The authors may wish to explain the mechanism behind increased rejection from 83 % (15 hours) to 99 % while maintaining the permeability at ~ 401 LMH/bar.

Reply:

We thank the reviewer for this constructive feedback and we apologize for this oversight. We have corrected the data in Supplementary Table 1. Moreover, we have repeated the performance evaluation of the membranes at 145 °C three times to ensure the reliability of the method and data. We have updated the table in the Supplementary Information file accordingly.

**Supplementary Table 1.** Molecular transport and separation evaluation of TFP-PDA membranes after specified time, prepared at various temperatures.

| Time (h) | 125 °C | | 135 °C | | 145 °C | | 155 °C | |
|---|---|---|---|---|---|---|---|---|
| | Permeance $(L \cdot m^{-2} \cdot h^{-}$ | CR rejection | Permeance $(L \cdot m^{-2} \cdot h^{-}$ | CR rejection | Permeance $(L \cdot m^{-2} \cdot h^{-}$ | CR rejection | Permeance $(L \cdot m^{-2} \cdot h^{-}$ | CR rejection |

	$^1 \cdot \text{bar}^{-1}$)	(%)	$^1 \cdot \text{bar}^{-1}$)	(%)	$^1 \cdot \text{bar}^{-1}$)	(%)	$^1 \cdot \text{bar}^{-1}$)	(%)
0	800 ±10	55±3	799±12	55±2	798±7	52±2	804±12	54±3
3	759±9	57±4	751±10	59±3	743±6	55±2	712±7	71±3
6	700±11	58±3	633±11	63±3	675±7	63±2	633±8	78±3
9	641±10	61±3	612±12	69±3	581±5	68±1	458±9	80±4
12	613±9	68±2	533±13	76±2	539±5	72±2	399±7	82±2
15	593±7	71±2	475±10	79±3	475±4	86±1	302±8	84±3
16	561±6	73±3	462±9	81±2	434±4	95±2	275±7	89±2
17	552±6	75±2	451±7	82±3	419±2	98±1	209±8	91±3
18	513±5	78±3	444±5	83±2	403±4	99±1	173±9	99±2
19	497±6	82±3	429±6	84±2	389±2	99±1	162±6	99±2
22	463±4	86±1	415±4	85±2	353±2	99±1	148±8	99±3

The optimization table for TFP-TTA is not provided anywhere in the paper.

Reply:

This is a very important point and we thank the reviewer for this gainful comment. We have provided the optimization Table for TFP-TTA in the revised Supplementary Information.

Supplementary Table 2. Molecular transport and separation evaluation of TFP-TTA membranes after specified time, prepared at various temperatures.

Time (h)	125 °C		135 °C		145 °C		155 °C	
	Permeance (L·m ⁻² ·h ⁻¹ ·bar ⁻¹)	CR rejection (%)	Permeance (L·m ⁻² ·h ⁻¹ ·bar ⁻¹)	CR rejection (%)	Permeance (L·m ⁻² ·h ⁻¹ ·bar ⁻¹)	CR rejection (%)	Permeance (L·m ⁻² ·h ⁻¹ ·bar ⁻¹)	CR rejection (%)
0	793 ±12	42±8	793±12	43±7	795±13	43±8	794±12	43±8
3	731±8	47±8	711±10	51±9	699±6	57±2	674±8	64±5
6	682±11	51±6	648±11	54±6	614±6	66±2	588±7	75±4
9	611±9	58±6	598±11	63±5	534±5	70±1	439±7	83±4
12	554±8	62±4	521±12	77±6	500±6	79±2	327±7	88±3
15	515±8	65±4	455±9	81±3	415±4	86±1	271±9	92±4

16	489±7	68±3	437±7	84±3	379±4	90±2	200±8	98±2
17	446±5	75±3	400±6	88±4	338±3	95±1	131±6	99±3
18	405±4	80±3	367±5	92±3	309±3	99±1	102±7	99±2
19	377±4	85±3	351±5	94±3	281±3	99±1	91±5	99±2
22	325±4	90±2	312±4	97±3	218±3	99±1	72±4	99±2

The authors may wish to explain the mechanism of the high rejection efficiency of Congo Red, which has a side that is small enough (~ 0.73 nm) to permeate through the large pores of TFP-PDA (1.4 nm).

Reply:

We thank the reviewer for this excellent and helpful advice. Congo red is a charged dye (anionic) with dimensions of 0.73 x 2.26 nm. In polar solvents such as water, it is important to consider the hydrodynamic radii of the solvated charged species which includes its solvation shell, thereby the larger dimension (2.26 nm) may be responsible for their high rejection rates but it could very well also be that the 0.73 nm dimension in solution has a much larger hydrodynamic cross section that cannot permeate through the pores. Additionally, dyes may also aggregate in water which substantially increases their overall dimensions. To make sure that this selectivity comes from the pores of COF membranes, we used PPH-IX dye with dimension of 1.45 x 1.54 nm in organic solvent and observed a rejection of more than 99%.

Also, the authors may consider giving a more detailed explanation of the low rejection and permeability at 155 °C as reference 38 does not seem to provide the information.

Reply:

We thank the reviewer for pointing out this issue. We optimized the temperature in the second step for COF membrane fabrication and observed that membranes crystallized at 155 °C exhibited poor permeance. We assume that COF particles grow in the pores which have also been observed by other groups as well (*Angew. Chem. Int. Ed.*, 2018, **57**, 4083-4087), eventually yielding membranes with low permeance. Secondly, high temperature treatment often rapidly leads to a kinetically stable state, but with poor orderliness. To confirm this, we characterized TFP-PDA membranes crystallized at 155 °C through XRD and BET. As expected, the crystallinity and surface area. of membranes crystallized at 155 °C are lower compared to those of crystallized at 145 °C. We felt justified in assuming that growth of COF particles in the pores and low crystallinity are the main causes of low

permeance for membranes prepared at 155 °C. Therefore, a temperature of 145 °C was determined as the optimum temperature with a rational balance between permeance and rejection.

We have revised the manuscript accordingly.

“At 155 °C, the low flux may arise from the partial blockage of pores by intergrown COF particles, also reported in previous literatures¹⁰. Secondly, treatment at high temperature leads to kinetically stable product rapidly, but with poor orderliness. To confirm this, we characterized TFP-PDA membranes crystallized at 155 °C through XRD and BET. Indeed, the crystallinity and surface area of membranes crystallized at 155 °C is lower than those of membranes crystallized at 145 °C. We assume that growth of COF particles in the pores and low crystallinity at 155 °C render membranes low permeance (Supplementary Fig. 8).”

We also included an additional figure in the Supplementary Information file.

Supplementary Fig. 8. XRD and BET surface area of TFP-PDA membranes pre-assembled at 60 °C (left) assembled at 145 °C, (right) assembled at 155 °C.

In general, this work is not particularly novel and presents a very similar structure as the authors' previous paper. It may not be suitable for Nature Communications.

We thank the reviewer for presenting us with this valuable opinion of our work. We would like to mention however, that the fabrication of COF powders and the fabrication of COF membranes differ significantly, as explained in our reply to this reviewer's first remark/question.

The motivation of this work was based on decoupling the polymerization and crystallization steps of the process and replacing liquids with vapors during crystallization as an efficient alternative to existing methods. We would like to clarify that the idea in this work is very innovative and as we have pointed out in our rebuttal, unprecedented, aiming to provide a generic, efficient, facile and broad strategy to fabricate COF membranes. We designed a phase-switching (switching from liquid phase to vapor phase) strategy to decouple polymerization and crystallization steps for the first-time during COF membrane fabrication. Moreover, the crystallization step was achieved using the vapor phase instead of the liquid phase. In our previous work of *J. Am. Chem. Soc.*, 2020, **142**, 31, 13450-13458, we exhibited a solid-vapor interfacial polymerization/crystallization method to obtain highly crystalline and ultra-thin membranes with inherent restrictions related to the melting point (M. P.) of vapor-phase monomers which should be below the reaction temperature. Moreover, in our previous work, we used PDA as vapor phase monomer (M.P. of PDA is ≈ 139 °C) whereas the reaction temperature was 150 °C). However, the M. P. of TTA monomer reported in this work is ≈ 365 °C, which would be impossible to be used in our previous work due to the too slow evaporation rate. Similarly, we also demonstrated that polymeric membranes could be transformed to crystalline membranes through monomer exchange strategy which took place in liquid phase (*Angew. Chem. Int. Ed.*, 2021, **60**, 33, 18051-18058). A polymeric membrane was first fabricated and transferred to a solvent system containing the exchange monomer. After 72 h, the monomer exchange was accomplished to produce crystalline monomers. However, large COF pores (>2.5 nm) are required to accomplish the exchange process, which is not suitable for nanofiltration separations. Thus, the novelty of this work differs significantly from previous reports in which vapors were used as structure-directing agents.

Finally, the previous reports as mentioned by this reviewer in the first remark/question (*J. Am. Chem. Soc.*, 2015, **137**, 3, 1016-1019; *Angew. Chem. Int. Ed.*, 2010, **50**, 3, 672-675; *Thin Solid Films*, 2013, **529**, 327-332; *J. Mater. Chem. A*, 2014, **2**, 8201-8204) are all based on either a one-step procedure or with no clear boundary between the polymerization and crystallization steps; the products are powders which are very difficult, even impossible in some cases, to process into continuous membranes. In this work, we present an alternative strategy to fabricate highly crystalline COF membranes through a two-step (phase-switching) procedure, free of all the specific restrictions for amine and aldehyde monomers that were present in all of the mentioned references by the reviewer.

Reviewer #3 (Remarks to the Author):

In this work, the authors report a phase-switching strategy to fabricate covalent organic framework (COF) membranes through a two-step procedure. The first step in liquid phase is primarily responsible for polymerization and the second step in vapor phase is primarily responsible for crystallization, which delicately decouples polymerization process and crystallization process. The two-step procedure has some distinct superiority to the commonly used one-step procedure in fabricating defect-free and ultrathin membranes. Using phase-switching strategy, highly crystalline, more compact and continuous membranes were fabricated. Accordingly, ultrafast molecular transport was realized, as manifested by water permeance of 402 L m⁻² h⁻¹ bar⁻¹ and acetonitrile permeance of 519 L m⁻² h⁻¹ bar⁻¹, which is among the ever-reported highest separation performance for COF membranes. The phase-switching strategy was validated by fabricating two kinds of highly crystalline COF membranes with different pore apertures. The strategy in this work represents a striking breakthrough in the precise construction of advanced COF membranes. This paper is meticulously organized and elegantly written and I read it with great joy. I firmly believe this interesting and innovative work will be highly appreciated by broad scientific communities in chemistry, materials science and membrane technology. Overall, this work is highly suitable for publication in Nature Communications after addressing the following minor revisions.

Reply:

We thank the reviewer for these highly positive remarks and valuable guidance.

1. The authors employed the β -ketoenamine-linked Schiff base COF in their work. More explanations may be required for the readers to more clearly understand why the authors preferred this type of COF over the other types?

Reply:

We thank the reviewer for this valuable guidance. In this work β -ketoenamine-linked Schiff base COFs were chosen primarily due to two reasons: (I) the initial reversible Schiff base reaction yields an enol which leads to crystalline arrangements which is subsequently converted to a ketone-enol tautomerized form; and (II) their excellent chemical stability under harsh acidic and basic conditions (*Small*, 2021, **17**, 2101017; *J. Am. Chem. Soc.*, 2017, **139**, 1856-1862). These features are important for our strategy as well as potential applications of membranes in practical conditions. Some COFs such as boronated COFs are highly sensitive to even small amounts of moisture, hindering their application in aqueous environments. To better convey this, we have updated the text in the revised manuscript accordingly.

“The initial reversible Schiff base reaction yields an enol form leading to crystalline arrangement which is subsequently converted to keto-enol tautomer form. Moreover, these COFs exhibit excellent chemical stability under harsh acidic and basic conditions^{11, 12}, unlike the boronated COFs, which are highly sensitive to even small amount of moisture and thus difficult to be utilized in aqueous environments¹³.”

2. Can the phase-switching strategy be applied to other kinds of COF membranes in principle?

Reply:

We thank the reviewer for this valuable guidance. We fabricated COF-LZU1 membranes based on imine linkage to further confirm its feasibility. The membranes crystallinity was confirmed by XRD and the completion of reaction was verified by FT-IR. We have modified the manuscript and Supplementary Information files accordingly.

“The reliability of our two-step procedure via phase-switching was further confirmed by the fabrication of COF-LZU1 membranes with defect-free surface and high crystallinity as evident from the XRD data with characteristic peaks similar to the literature¹⁴. The completion of reaction was confirmed through FT-IR with the disappearance of peaks from the initial monomers and the formation of imine bonding (Supplementary Fig. 9).

Supplementary Fig. 9. (a) COF-LZU1 chemical structure, (b) SEM exhibiting large defect-free area of the membrane, (c) XRD of the pristine and COF-LZU1 membranes, (d) FT-IR spectra of 1,4-diaminobenzene (black), 1,3,5-triformylbenzene (red), and COF-LZU1 membrane (blue).

Moreover, the method section was also modified accordingly, as shown below.

“Methods

Pre-assembly step in liquid phase toward pristine membranes

TFP (2.10 mg, 0.01 mmol), 1,3,5-triformylbenzene (TFB, 1.62 mg, 0.01 mmol), PDA (1.62 mg),”

“Next, the mixed solution containing TFP/PDA, TFP/TTA or TFB/PDA was poured on Indium Tin Oxide”

3. Why did the authors choose DMAc as the solvent in the first step? Can the authors outline the major criteria in choosing the solvent?

Reply:

We would like to thank the reviewer for these constructive questions. The major reasons for choosing DMAc are the high solubility of amine and aldehyde monomers in DMAc as well as the appropriately slow rate of evaporation for completion of the polymerization in the first step. Due to the monomer's high solubility (TFP \approx 23mg/ml, PDA \approx 25mg/ml and TTA \approx 17 mg/ml), a low volume of DMAc is required in the first step. Secondly, due to the relatively high boiling point of DMAc, it evaporates slowly at 60 °C allowing the completion of polymerization. In our experiments, we observed that 2 ml of DMAc requires about 25 minutes for its evaporation at 60 °C. When a low boiling point solvent such as dichloromethane is used, the solvent evaporates very fast leaving no sufficient time for the completion of the polymerization in the first step. We have added this text to the revised supplementary Information file.

“DMAc was chosen in this step due to high solubility of amine and aldehyde monomers in DMAc which reduces reaction volumes (TFP \approx 23mg/ml, PDA \approx 25mg/ml and TTA \approx 17 mg/ml) and relatively high boiling point of DMAc, which wins sufficient evaporation time for the completion of the polymerization at this step.”

4. Have the authors tested the rejection performance of dyes in organic solvents?

Reply:

We would like to thank the reviewer for another valuable question. We have tested the dye, PPH-IX, in organic solvent to make sure that the membrane's pores are broadly suitable for dye rejections. The text in the manuscript was revised accordingly.

“Dyes can form aggregates in aqueous solution at high concentrations^{15, 16}, therefore, an ethanolic solution of PPH-IX was used to confirm the broad applicability of COF membranes for the dyes rejection¹⁷.”

Some mistakes:

(1) The scale bar of cross-section SEM in Fig. 2c, f should be corrected.

Reply:

We express thanks to the reviewer for catching this mistake and apologize for this oversight. We have corrected the mistake in the revised manuscript.

Fig. 2. Structural characterization: (a, d) PXRD pattern, experimental (black line) and simulated (red line), digital photos of the COF membranes on the support are also shown. (b, e) HR-TEM images and SAED patterns. (c, f) Surface and cross-section SEM of the TFP-PDA, TFP-TTA respectively.

(2) In line 125, XRD was written twice.

Reply:

We appreciate the reviewer's watchful eye in catching this error. We have corrected the error in the revised manuscript.

“*via* phase switching was evaluated through FTIR, XRD, X-ray photoelectron spectroscopy (XPS),”

In addition, we would like to make some revisions on the author affiliations and acknowledgements, as shown below.

In the author affiliations part, we added “Haihe Laboratory of Sustainable Chemical Transformations, Tianjin 300192, China” due to its financial support and removed “Joint School of National University of Singapore and Tianjin University, International Campus of Tianjin University, Binhai New City, Fuzhou 350207, China” and changed the order of “Zhejiang Institute of Tianjin University, Ningbo, Zhejiang 315201, China”, according to their contributions.

The manuscript was revised as follows:

“Niaz Ali Khan^{1,2}, Runnan Zhang^{1,2,5*}, Xiaoyao Wang^{1,2}, Li Cao¹, Chandra S. Azad⁶, Chunyang Fan^{1,2}, Jinqiu Yuan^{1,2}, Mengying Long^{1,2}, Hong Wu^{1,2,3*}, Mark. A. Olson⁷ and Zhongyi Jiang^{1,2,4,5*}

¹Key Laboratory for Green Chemical Technology of Ministry of Education, School of Chemical Engineering and Technology, Tianjin University, Tianjin 300072, China.

²Collaborative Innovation Center of Chemical Science and Engineering (Tianjin), Tianjin 300072, China.

³Tianjin Key Laboratory of Membrane Science and Desalination Technology, Tianjin University, Tianjin 300072, China.

⁴Joint School of National University of Singapore and Tianjin University, International Campus of Tianjin University, Binhai New City, Fuzhou 350207, China

⁵Zhejiang Institute of Tianjin University, Ningbo, Zhejiang 315201, China.

⁶Department of Chemistry, Northwestern University, 2145 Sheridan Rd., Evanston, IL 60208 USA

⁷Department of Physical and Environmental Sciences, Texas A&M University Corpus Christi, 6300 Ocean Dr., Corpus Christi, TX 78412 USA”

was revised as

“Niaz Ali Khan^{1,4}, Runnan Zhang^{1,2,3,4*}, Xiaoyao Wang^{1,4}, Li Cao¹, Chandra S. Azad⁶, Chunyang Fan^{1,4}, Jinqiu Yuan^{1,4}, Mengying Long^{1,4}, Hong Wu^{1,3,4,5*}, Mark. A. Olson⁷ and Zhongyi Jiang^{1,2,3,4*}

¹Key Laboratory for Green Chemical Technology of Ministry of Education, School of Chemical Engineering and Technology, Tianjin University, Tianjin 300072, China.

²Zhejiang Institute of Tianjin University, Ningbo, Zhejiang 315201, China.

³Haihe Laboratory of Sustainable Chemical Transformations, Tianjin 300192, China.

⁴Collaborative Innovation Center of Chemical Science and Engineering (Tianjin), Tianjin 300072, China.

⁵Tianjin Key Laboratory of Membrane Science and Desalination Technology, Tianjin University, Tianjin 300072, China.

⁶Department of Chemistry, Northwestern University, 2145 Sheridan Rd., Evanston, IL 60208 USA

⁷Department of Physical and Environmental Sciences, Texas A&M University Corpus Christi, 6300 Ocean Dr., Corpus Christi, TX 78412 USA”

In the acknowledgements part, we added “Haihe Laboratory of Sustainable Chemical Transformations” due to its financial support during revision.. The manuscript was revised as follows: “This project was supported by Key Research and Development Program of Zhejiang Province (2021C03173), National Natural Science Foundation of China (91934302, 21961142013, 22008172), Research Fund for International Young Scientists funded by NSFC (21850410457), Project funded by China Postdoctoral Science Foundation (2020TQ0226, 2021M692384) and Program of Introducing Talents of Discipline to Universities (BP0618007). We also thank the Haihe Laboratory of Sustainable Chemical Transformations for financial support. We acknowledge the help of Y. Chao during figures drawing”

1. Yang J, *et al.* Protonated Imine-Linked Covalent Organic Frameworks for Photocatalytic Hydrogen Evolution. *Angew. Chem. Int. Ed.* **60**, 19797-19803 (2021).
2. Wang XF, *et al.* Particle Dissolution and Recrystallization Progress of Al-Mg-Si-Cu Alloy during Solution Treatment. *Phys. Met. Metallogr.* **121**, 1258-1265 (2020).
3. Inoue A, Utada M, Hatta T. Halloysite-to-kaolinite transformation by dissolution and recrystallization during weathering of crystalline rocks. *Clay Miner.* **47**, 373-390 (2012).
4. Wang YR, Lin M, Tuel A. Hollow TS-1 crystals formed via a dissolution-recrystallization process. *Microporous Mesoporous Mater.* **102**, 80-85 (2007).
5. Tao HX, Ren JW, Liu XH, Wang YQ, Lu GZ. Facile synthesis of hollow zeolite microspheres through dissolution-recrystallization procedure in the presence of organosilanes. *J. Solid State Chem.* **200**, 179-188 (2013).
6. Tu M, Wannapaiboon S, Fischer RA. Inter-conversion between zeolitic imidazolate frameworks: a dissolution-recrystallization process. *J. Mater. Chem. A.* **8**, 13710-13717 (2020).
7. Chen X, Zhang H, Zhang MK, Zou Y, Zhang S, Qu YQ. Temperature-responsive dissolution/recrystallization of Zn MOF enables the maximum efficiency and recyclability of catalysts. *Chem. Commun.* **56**, 1960-1963 (2020).
8. Sun YT, Ding S, Xu SS, Duan JJ, Chen S. Metallic two-dimensional metal-organic framework arrays for ultrafast water splitting. *J. Power Sources* **494**, 229733 (2021).
9. Dey K, *et al.* Selective molecular separation by interfacially crystallized covalent organic framework thin films. *J. Am. Chem. Soc.* **139**, 13083-13091 (2017).
10. Fan H, Gu J, Meng H, Knebel A, Caro J. High-flux membranes based on the covalent organic framework COF-LZU1 for selective dye separation by nanofiltration. *Angew. Chem. Int. Ed.* **57**, 4083-4087 (2018).
11. Wang L, *et al.* Activation of Carbonyl Oxygen Sites in β -Ketoenamine-Linked Covalent Organic Frameworks via Cyano Conjugation for Efficient Photocatalytic Hydrogen Evolution. *small* **17**, 2101017 (2021).
12. Karak S, *et al.* Constructing Ultraporous Covalent Organic Frameworks in Seconds via an Organic Terracotta Process. *J. Am. Chem. Soc.* **139**, 1856-1862 (2017).

13. Côté AP, Benin AI, Ockwig NW, O'Keeffe M, Matzger AJ, Yaghi OM. Porous, Crystalline, Covalent Organic Frameworks. *Science* **310**, 1166-1170 (2005).
14. Ding S-Y, *et al.* Construction of Covalent Organic Framework for Catalysis: Pd/COF-LZU1 in Suzuki–Miyaura Coupling Reaction. *J. Am. Chem. Soc.* **133**, 19816-19822 (2011).
15. Coates E. Aggregation of Dyes in Aqueous Solutions. *Journal of the Society of Dyers and Colourists* **85**, 355-368 (1969).
16. Navarro A, Sanz F. Dye aggregation in solution: study of CI direct red I. *Dyes and Pigments* **40**, 131-139 (1999).
17. He X, *et al.* Controlling the Selectivity of Conjugated Microporous Polymer Membrane for Efficient Organic Solvent Nanofiltration. *Adv. Funct. Mater.* **29**, (2019).

Reviewers' Comments:

Reviewer #1:

Remarks to the Author:

I am satisfied with the revision and recommend the publication.

Reviewer #3:

Remarks to the Author:

The authors have well addressed the comments of reviewer #2 and #3, and this manuscript has been greatly improved after the revision, could be accepted for publication.

REVIEWER COMMENTS

Reviewer #1 (Remarks to the Author):

I am satisfied with the revision and recommend the publication.

Reply:

Thanks to the reviewer for the highly positive remarks.

Reviewer #3 (Remarks to the Author):

The authors have well addressed the comments of reviewer #2 and #3, and this manuscript has been greatly improved after the revision, could be accepted for publication.

Reply:

Thank the reviewer for the highly positive comments and the efforts in reviewing this manuscript.